# Relationships between light exposure and aspects of cognitive function in everyday life
Altug Didikoglu [1,2] ✉, Tom Woelders[2], Lucien Bickerstaff [3,4], Navid Mohammadian[5], Sheena Johnson[6], Martie van Tongeren[7], Alexander J. Casson[5], Timothy M. Brown [8] & Robert J. Lucas [2] ✉

Light exposure can modulate cognitive function, yet its effects outside of controlled laboratory settings remain insufficiently explored. To examine the relationship between real-world light exposure and cognitive performance, we assessed personal light exposure and measured subjective sleepiness, vigilance, working memory, and visual search performance over 7 days of daily life, in a convenience sample of UK adults ($n$ = 58) without significant circadian challenge (shiftwork or jet-lag). A subset of participants ($n$ = 41) attended an in-lab session comprising a battery of pupillometric and psychophysical tests aimed to quantify melanopsin-driven visual responses. We find significant associations between recent light exposure and subjective sleepiness. Recent light exposure was also associated with reaction times for both psychomotor vigilance and working memory tasks. In addition, higher daytime light exposure and an exposure pattern with reduced fragmentation were linked to improved cognitive performance across visual search, psychomotor vigilance, and working memory tasks. Higher daytime light exposure and earlier estimated bedtimes were associated with stronger relationships between recent light exposure and subjective sleepiness. These results provide real world support for the notion that intra- and inter-individual differences in light exposure meaningfully influence aspects of cognition, with beneficial effects of short-term bright light and of habitual light exposure patterns characterized by brighter daytimes, earlier rest phase, and greater intra- and inter-daily stability.

Light is a fundamental environmental cue that governs numerous biological processes in humans, including circadian rhythms, sleep, and cognition[1–3]. The discovery of intrinsically photosensitive retinal ganglion cells (ipRGCs) has significantly advanced our understanding of how light influences these processes beyond vision[4–6]. These ipRGCs play a central role in regulating the circadian system by modulating circulating melatonin, setting circadian phase, and alertness[7–9]. The relationship between light exposure and cognitive performance has garnered significant attention in recent years, with

direct retinal pathways identified as key regulators of cognitive function of animal models[4]. Furthermore, high-resolution brain imaging of humans has revealed that light enhances cognitive functions, though its effects are complex, eliciting distinct responses in specific light-sensitive subnuclei of the hypothalamus and varying across different cognitive tasks[10–12].

Laboratory studies have demonstrated that exposure to bright light during both day and night can acutely enhance cognitive performance[13–22]. On the other hand, nighttime light exposure disrupts sleep, leading to

[1]Department of Neuroscience, Izmir Institute of Technology, Gulbahce, Urla, Izmir, Turkey. [2]Centre for Biological Timing, Division of Neuroscience, School of Biological Sciences, Faculty of Biology Medicine and Health, University of Manchester, Manchester, UK. [3]Max Planck Institute for Biological Cybernetics, Translational Sensory & Circadian Neuroscience, Tübingen, Germany. [4]TUM School of Medicine and Health, Technical University of Munich, Munich, Germany. [5]Department of Electrical & Electronic Engineering, School of Engineering, Faculty of Science and Engineering, University of Manchester, Manchester, UK. [6]Thomas Ashton Institute, People, Management and Organisation Division, Alliance Manchester Business School, Faculty of Humanities, University of Manchester, Manchester, UK. [7]Thomas Ashton Institute, Centre for Occupational and Environmental Health, Division of Population Health, Health Services Research & Primary Care, School of Health Sciences, Faculty of Biology Medicine and Health, University of Manchester, Manchester, UK. [8]Centre for Biological Timing, Division of Diabetes Endocrinology and Gastroenterology, School of Medical Sciences, Faculty of Biology Medicine and Health, University of Manchester, Manchester, UK. ✉e-mail: altugdidikoglu@iyte.edu.tr; robert.lucas@manchester.ac.uk

cognitive impairments the following day[7,23–25]. Adding further complexity, cognitive functions themselves follow circadian rhythms[1,26,27]. Additionally, individual differences contribute to variations in how light influences cognition[28,29]. Despite substantial findings from controlled laboratory studies, a critical gap persists in understanding how these effects translate to real-world environments, where light exposure is dynamic and intertwined with daily routines.

Bridging this gap is essential for comprehending light's broader implications on daily cognitive performance. Real-world light exposure varies widely in its stability, intensity, timing, and spectral composition due to the interplay of natural and artificial light sources. In modern industrialized societies, where individuals spend most of their time indoors and are frequently exposed to artificial lighting and night-time light, understanding the cognitive effects of light exposure is crucial to enhance workplace efficiency, to improve health and safety, and to support better educational outcomes[30–36]. Furthermore, these investigations illuminate potential pathways for developing interventional therapies aimed at mitigating cognitive decline and dementia[37]. To address these gaps, this study investigates the impact of light exposure on cognitive performance in everyday life. We hypothesized that recent increases in light intensity would acutely enhance cognitive performance, that weekly light exposure patterns would influence general cognitive performance, and that individual photosensitivity of cognitive functions could be estimated through light exposure metrics or in-lab sensitivity assessments. Utilizing innovative tools – a wearable melanopic light monitor (Spectrawear[38]) and a mobile application (Brightertime[39]) – we collected continuous data on light exposure and cognitive performance over 7 days of everyday life. This approach enables an assessment of both the acute and cumulative effects of light exposure on cognition, and individual differences in photosensitivity. Our findings contribute to the growing body of evidence on light's role in modulating cognition and emphasize the importance of personalized strategies for optimizing light exposure to enhance cognitive performance and overall well-being.

## Methods

The study was not preregistered. The full study protocol was shared in protocols.io (https://doi.org/10.17504/protocols.io.n92ldrjjxg5b/v1).

### Participants

A total of 60 individuals participated in the study, conducted between July 2022 and August 2023 in Manchester, UK. To ensure the power of estimates, a minimum sample size of 50 participants was targeted[40]. Study weeks started on any day of the weekdays. Participants were eligible for inclusion if they were at least 18 years old, employed either full-time or part-time, had no history of intercontinental travel in the preceding two weeks, and had not been diagnosed with a sleep disorder. The study aimed to capture a real-world population; therefore, no specific health-related exclusion criteria were applied. While this approach increased ecological validity, it may have introduced limitations in detecting certain associations. Two participants were excluded from the final analyses due to study design noncompliance and insufficient data provision. The final sample consisted of 29 males and 29 females. Sex and gender were determined based on information provided by the participants. No race or ethnicity information was collected in this study. Age distribution was as follows: 14 participants were <25 years old, 23 were between 25 and 30, 13 were between 30 and 35, and the remaining participants were over 35 years old. The majority (n = 55) held a higher education degree, and 32 were employed full-time. Four participants occasionally engaged in shift work, with eight days reported as shift work during the study; however, none involved night shifts that significantly alter night sleep patterns. The dataset comprised 65% workdays. Regarding health characteristics, 46 participants were non-smokers, two were color-blind, three had an ADHD diagnosis, 11 reported anxiety or depression, and 3 reported poor overall health. Sleep quality, assessed via the Pittsburgh Sleep Quality Index (PSQI), indicated generally good sleep health, with a mean score of 6.4 (SD = 1.9). Chronotype was assessed using the Munich

Chronotype Questionnaire (MCTQ), with a mean midsleep time on free days (MSFsc) of 5:04 AM (SD = 1.4, range: 2:15 AM–7:51 AM), ensuring a diverse representation of sleep timing preferences.

### Study design and light monitoring

We employed a real-world protocol for monitoring personal light exposure[41,42]. This approach allowed us to assess natural variations in light exposure and its effects on cognition outside of controlled laboratory settings. Participants were asked to remain in the study for one week, though they had the flexibility to withdraw at any time or to extend their participation. Data collection involved two key technologies: 1) a wearable melanopic equivalent daylight exposure monitor for personal light measurements (Spectrawear[38]) and 2) a mobile application for cognitive task performance and questionnaire responses (Brightertime[39]). During the initial registration session, participants were provided with the light monitor, guided through account setup for the Brightertime app, and instructed on the proper usage of both technologies.

Spectrawear is a multichannel light sensor designed to measure melanopic equivalent daylight exposure. Above 1 lx, the device demonstrated an average mean absolute log deviation of < 0.05 log units, with a minimum between-device correlation of 0.99. Its typical performance estimates α-opic equivalent daylight illuminances (EDI) within ±15% across a wide dynamic range (1 to 100,000 lx). In addition, the angular response to incident light was with a half-width at half maximum of 51°. The device was worn on the non-dominant wrist and set to record light exposure at 30 s intervals. Participants were instructed to wear the device throughout the day and remove it just before bedtime, placing it in a consistent location within the same room (preferably near eye level). Participants were advised to charge it daily, preferably overnight. Participants did not have direct access to their recorded data, nor were they able to modify device settings. Additionally, they were asked to avoid wearing clothing that could obstruct the light sensor.

Throughout the study, participants used the Brightertime app to complete the following tasks: 1) baseline surveys administered during the registration meeting; 2) daily sleep and work diary completed each morning; 3) subjective sleepiness reports submitted at their discretion throughout the day, with a recommended schedule of morning, midday, and evening entries; 4) cognitive assessments, including the Psychomotor Vigilance Task, N-Back, and Visual Search Task, performed each time a subjective sleepiness report was submitted. Participants were encouraged to provide multiple recordings at various times of the day rather than limiting themselves to three fixed entries. To maintain ecological validity, participants were explicitly instructed to follow their normal routines and not alter their sleep or lighting behaviors during the study. At the conclusion of the study week, participants were offered an optional in-lab assessment of light sensitivity[43]. Those who accepted (n = 41) underwent a session evaluating melanopic brightness preference, subjective brightness, and pupillary light reflex responses.

### Surveys and measures

The surveys were administered at specified times through the Brightertime app[39]. The baseline questionnaire included a study-specific sociodemographic and health survey, along with standardized sleep and chronotype assessments such as the Pittsburgh Sleep Quality Index (PSQI)[44] and the Munich Chronotype Questionnaire (MCTQ)[45]. The sociodemographic and health survey collected data on age, sex, employment and shiftwork status, subjective health rating, prior diagnoses of sleep, eye, mental, or neurological disorders, as well as daily caffeine, alcohol, and smoking habits. Chronotype was quantified using the MCTQ-derived midsleep time on free days, adjusted for sleep debt on workdays (MSFsc). Sleep quality was assessed using the PSQI, with higher scores (out of 21) indicating greater sleep disturbances. The morning diary recorded whether it was a workday or a free day, bedtime, and wake time. Participants also reported their total sleep duration (in hours) and sleep latency (time taken to fall asleep in minutes). Repeated assessments of subjective sleepiness were conducted

using the 10-item version of the Karolinska Sleepiness Scale (KSS)[46], where a score of 10 indicated extreme sleepiness. Before each repeated assessment, if napping occurred, the time of the last awakening was recorded.

## Cognitive tasks

The cognitive tasks used in this study were previously detailed in the Brightertime development article[39]; however, the number of trials was further optimized to reduce task duration and trial numbers to a minimum, while preserving the mean and variance of the results. Participants used their own smartphones and were eligible if they had a functional, non-cracked touchscreen. Upon first logging into the app, participants were required to complete a short practice session for each cognitive task to minimize learning effects. At the end of the practice, they could either repeat the session or proceed with the study. Each cognitive session comprised three tasks: the Psychomotor Vigilance Task (PVT), N-Back ($n = 3$; NB3), and Visual Search Task (VS). To mitigate order effects, the app randomized the task presentation sequence in each session. Response accuracy (correct hits, correct rejections, misses, and false alarms) and reaction times were recorded, and results were synchronized to a study-specific server upon session completion via an internet connection.

The Psychomotor Vigilance Task assessed sustained attention by requiring participants to monitor a central target and tap the screen as quickly as possible upon the appearance of a stimulus (a cartoon zombie). The task consisted of 28 stimulus presentations with inter-stimulus intervals randomly set between 2 and 10 s and a stimulus time-out of 1 s. The working memory task evaluated short-term memory, where participants monitored a sequence of nine letters (A, B, D, E, K, M, R, S, T) and responded when the presented letter matched the one from three trials prior. The task included 22 letter presentations with six targets and a stimulus time-out of 2 s. The visual search task measured visual cognitive performance, requiring participants to search for a monkey (with a "T"-shaped nose) among human distractors (with an "L"-shaped nose) displayed in one of four orientations. Difficulty was varied by randomly presenting 12, 24, or 36 figures. The task comprised 42 stimulus presentations with a random present/absent ratio (minimum 0.3, maximum 0.8, median 0.5), a 2 s inter-stimulus interval, and a stimulus time-out of 10 s.

Incomplete data due to app crashes (the app then restarts) were excluded, along with reaction times below 100 ms to filter out invalid responses. From the raw data, key performance metrics were computed for each task. For Psychomotor Vigilance Task, these included false negative rate (FNR, %), false discovery rate (FDR, %), accuracy (ACC, % of hits and correct rejections among trials), median reaction time of hits (median RT, ms), inverse efficiency score (IES; mean reaction time/ACC), 90th percentile of reaction times (slow10, ms), 10th percentile of reaction times (fast10, ms), and lapses (number of responses above 500 ms). For working memory, the computed variables were FNR, false positive rate (FPR, %), FDR, false omission rate (FOR, %), ACC, discrimination index (d-prime), median RT, IES, slow10, and fast10. For visual search, the calculated metrics included FNR, FPR, FDR, FOR, ACC, d-prime, median RT, IES, slow10, fast10, and search efficiency slopes (linear regression slope of reaction time across search density). Participants used their own devices, which may have introduced differences in touchscreen sensitivity. However, this issue was addressed and found to have no effect on cognitive task scores[39]. Additionally, data cleaning procedures excluded sessions with low accuracy (< 30%), specifically five sessions for working memory, three for Psychomotor Vigilance Task, and two for visual search. Furthermore, two participants were removed from the analysis for having fewer than seven valid task entries per game over the study week.

The learning effects of cognitive outcomes were investigated (see Fig. S1 for details). Overall, subjective sleepiness and Psychomotor Vigilance Task were independent of the number of tasks completed per day and showed no learning or motivation effect. In contrast, working memory and visual search, which were slightly more complex tasks, exhibited a learning effect, with participants becoming more accurate and faster the longer they remained in the study. Increased task repetition led to greater practice

effects, though this was only observed in visual search IES. Any potential confounding effects of learning were controlled for, as described in the statistical analysis.

## In-lab light sensitivity assessment

The in-lab light sensitivity measurement equipment and protocol were previously published[43]. A multiprimary projector system was used to conduct silent substitution experiments or deliver single-color light stimuli. Participants were dark-adapted for five minutes before the procedure. Following a calibration step involving a flicker photometry task (7.5 Hz) to match the luminance of two isoluminant spectra differing only in melanopic irradiance. Participants then completed three assessments: melanopic brightness discrimination, subjective brightness evaluation, and post-illumination pupil response (PIPR) measurement. In the melanopic brightness preference task, participants were shown 80 trials of spectral pairs, one without melanopic contrast and the other with varying contrasts (0–53% Michelson contrast, 20 steps, four repeats per step). They were asked to select the spectrum that appeared brighter. Two parameters were derived: the proportion of trials where the spectrum with higher melanopic irradiance was perceived as brighter at 53% contrast and the contrast level at which it was selected as brighter in 75% of trials. For subjective brightness assessment, participants viewed a high-melanopic-irradiance spectrum and rated its brightness on a 1–100 scale, with reference points at 0 (completely dark), 25 (dim), 50 (just bright and comfortable), 75 (bright), and 100 (too bright and uncomfortable). The PIPR assessment involved an initial 10 min dark adaptation, followed by 1 s flashes of red or blue light in random order. Each flash was preceded by 10 s and followed by 30 s of dark adaptation. The initial pupil constriction was quantified as the maximal initial pupil reflex to the blue stimulus, expressed as a percentage of the baseline diameter (mean over 10 s before the flash). The PIPR response was calculated as the area under the curve (AUC) for baseline-subtracted pupil diameter over the 0–6 s interval after light offset. Using the red stimulus as a reference, the difference between blue and red AUC values represented the PIPR for each participant. Additionally, a normalized PIPR amplitude was calculated as a percentage of the initial constriction amplitude.

## Statistical analysis

All data processing and analyses were conducted using R version 4.4.1 (2024) and MATLAB version R2023b, with data visualization performed in GraphPad Prism version 10.3.0. Melanopic lux values were $\log_{10}$-normalized for all calculations and summaries. In each section, we applied the Bonferroni correction to adjust for multiple comparisons by setting the significance threshold as α divided by the number of predictors. Data distribution was assumed to be normal, but this was not formally tested. Two-tailed hypothesis tests were used. Descriptive variables were reported as means and standard deviations (SD). Bivariate comparisons were conducted using Pearson correlation coefficients and independent-samples t-tests.

To determine the primary cognitive outcome variables, we performed factor analysis. Principal Component Analysis (PCA) was used to extract eigenvalues, and factors were selected based on eigenvalues greater than 1. Factor analysis was then conducted using the selected number of factors with varimax rotation, and variables were assigned to factors based on their highest loadings. This analysis identified key cognitive measures: subjective sleepiness, two factors for Psychomotor Vigilance Task (median RT and accuracy), three factors for working memory (median RT, FPR, and FNR), and three factors for visual search (IES, FPR, and FNR).

Cognitive performance outcomes were described using distributions across different times of the day (clock time) and time awake (duration between the last wake time and the time of cognitive tasks). For all cognitive measures, linear mixed models (LMM) with random slopes and intercepts for subjects were used to examine associations with time of day, time awake, and previous night's sleep duration and sleep midpoint time. Time awake was modelled as a second-degree polynomial function, while time of day was represented using a harmonic fit with sine and cosine terms [$\sin(2\pi \times$ time

of day/24) + cos(2π × time of day/24)]. First-degree functions were used to model the duration of the previous night's sleep (hours) and the sleep midpoint. The standardized coefficients were visualized using a heatmap for comparison. We standardized all fixed-effect coefficients by rescaling each predictor by its sample SD and the outcome by its SD, yielding interpretable standardized coefficients with 95% confidence intervals. We then conducted an ANOVA on the fitted linear mixed model to test each fixed effect's significance and converted each F-statistic (with its degrees of freedom) into a partial $\eta^2$ effect size. In addition, to investigate the learning effect and the influence of the number of tasks completed per day, we fit a separate linear mixed-effects model for each cognitive outcome, using experiment day and total tasks per day as predictors.

To examine whether cognitive task performance was correlated with recent light exposure, we calculated the mean light exposure over the prior 30, 60, 90, and 120 min. We then compared these mean values of light exposure with cognitive outcomes, adjusting for time of day and time awake, and the previous night's sleep duration and midpoint. Cognitive outcomes were modelled using LMM, with random slopes and intercepts for light exposure specific to each individual. The standardized coefficients were visualized using a heatmap for comparison. For working memory, the model further adjusted for task day to account for the learning effect. We retained the individual linear regression slopes from the models of subjective sleepiness, Psychomotor Vigilance Task, and working memory vs. 30-minute light history as a measure of real-world light sensitivity in cognition.

Pearson correlations were used to compare the relationships between real-world light sensitivity of cognition, in-lab photosensitivity variables, weekly light exposure variables, and weekly cognitive output means. Weekly cognitive output averages were calculated as the means across the week. Weekly light exposure predictors include intensity variables such as M10 (the mean melanopic EDI of the brightest 10-hour period) and L5 (the mean melanopic EDI of the dimmest 5 h period); duration variables such as the time spent above 250 Melanopic EDI lux (in minutes) and the time spent above 10 Melanopic EDI lux after sunset (in minutes); timing variables including the midpoint time of the M10 period, the midpoint time of the L5 period, and the last time above 1 Melanopic EDI lux; and stability variables such as IS (interdaily stability of light exposure) and IV (intradaily variability of light exposure). The IS of melanopic EDI was calculated as a metric of the similarity of daily light exposure patterns during the study period by comparing the variance of average hourly means to the variance of all hourly means. The IV of melanopic EDI was calculated to assess the fragmentation of light exposure patterns during the study period by comparing the mean square deviation of hourly means from the previous hour to the variance of all hourly means. If a correlation remained significant after correction for multiple testing, the predictor and outcome were further compared using linear regression, adjusted for age, sex, daylength, caffeine consumption, alcohol consumption, smoking, chronotype (MSFsc), and sleep problems (PSQI).

### Ethics
This project was approved by the University of Manchester Research Ethics Committee (Ref: 2021-12948-20856 and Ref: 2023-16080-26819). All participants provided informed consent before commencing the study.

### Reporting summary
Further information on research design is available in the Nature Portfolio Reporting Summary linked to this article.

## Results
### Measuring cognitive performance in real world settings
We aimed to collect cognitive function data in everyday life using an accessible smartphone app, Brightertime[39]. This app incorporates a Psychomotor Vigilance Task (PVT) to measure sustained attention, an N-back task (NB3; 3-back) to assess working memory, a T vs. L visual search task (VS) to evaluate search accuracy and efficiency, and subjective sleepiness report (KSS) (Fig. 1A). Participants (n = 58) used the app multiple times

throughout the day at their discretion (Fig. 1B). (Fig. S2: min = 13, max = 52, median = 23 entries per participant over an 8-day study period; Fig. S3: a representative study design for one participant). Across the 1428 subjective sleepiness reports, we captured the full range of subjective sleepiness, indicating that our methodology appropriately captured real-world variations in 'tiredness' throughout the day. The data exhibited minor positive skewness (0.4), with a minimum value of 1, a maximum value of 10, a mean of 4.5, and a standard deviation of 1.9.

Objective tasks of vigilance (n = 1334), working memory (n = 1341), and visual search (n = 1340) recorded correct hits, misses, false alarms, and reaction times for each attempt (ms). Using these records, a set of cognitive variables was extracted, including median reaction times of hits, accuracy, false negative/positive rates, difficulty slope of visual search, number of lapses in vigilance task (>500 ms), discriminability score (d') of working memory and visual search, and inverse efficiency slopes (IES; mean reaction times per ratio of correct answers). The summary statistics of the cognitive measures were comparable to previous real-world cognitive task findings[39] (Table S1). The average of the median reaction times for the vigilance task was 421.5 ms (min = 263.0 ms, max = 800 ms, SD = 79.3), with an average accuracy of 93.8% (SD = 9.4) and an average number of lapses of 6.1 (SD = 5.8). For the working memory task, the average of the median reaction times to correctly remember a letter within a 2 s window was 750.5 ms (min = 285.5 ms, max = 1755.0 ms, SD = 234.3 ms), with an average accuracy of 84.8% (SD = 12.6) and an average d' of 2.0 (SD = 0.9). In the visual search task, the average of the median reaction times to correctly answer within a 10-s window was 2637.4 ms (min = 295.0 ms, max = 7523.5 ms, SD = 821.9 ms), with an average accuracy of 87.9% (SD = 9.6) and an average d' of 2.7 (SD = 0.7).

Since each objective task had multiple outcome variables that were commonly correlated with each other (Pearson correlation up to 0.96), we performed dimension reduction using factor analysis (Table S2). This analysis resulted in two factors for vigilance (median reaction time and accuracy), three factors for working memory (median reaction time, false positive rate, and false negative rate), and three factors for visual search (inverse efficiency score, false positive rate, and false negative rate) (Fig. S4). The remaining analysis will therefore assess these variables.

### Rhythm of cognitive performance in everyday life
As expected, light exposure assessed with the wearable melanopic light logger (Spectrawear[38]) (Fig. 1A) exhibited time of day dependence. Across all 497 days of recording, global mean irradiance was 0.9 log lux melanopic equivalent daylight illumination (EDI) (SD = 1.5) (Fig. 2A). The maximum melanopic EDI per subject ranged from 4.4 to 5.2 log lux.

It is plausible that the cognitive variables we measured exhibit a diurnal rhythm, influenced by the natural circadian cycle. Furthermore, the timing and duration of recent sleep episodes, as well as the interval since the last waking, may significantly influence these cognitive outcomes. Therefore, we continued our analysis by investigating the rhythmic properties of cognitive function. The median time of day when cognitive tasks were performed was 16:06, with all clock times except 4–6 AM represented in our dataset. There was no statistically significant association between the median task completion time for each subject and the number of tasks completed during the week (r(56) = 0.08, 95% CI [–0.17, 0.33], p = 0.57), indicating that task timing preferences were not biased by the number of tasks completed. The mean time awake prior to task performance was 7.6 h (SD = 5.3). Additionally, the mean duration of the last sleep episode before each task was 7.1 h (SD = 1.2). The midpoint of the last sleep period occurred at an average time of 4:29 AM (SD = 1.4 h).

Subjective sleepiness scores showed strong associations with time of day, time awake, and the duration of the previous sleep episode (Fig. 2B). The subjective sleepiness model revealed that subjective sleepiness was typically higher within the first hour after wakening (mean=4.9) than later in the wake episode (mean daytime minimum = 3.7), indicative of sleep inertia, and then increased at approximately 0.4 subjective sleepiness points per hour later in the day (Time awake Std. Coef. = –1.04, 95% CI [–1.33, –0.75],

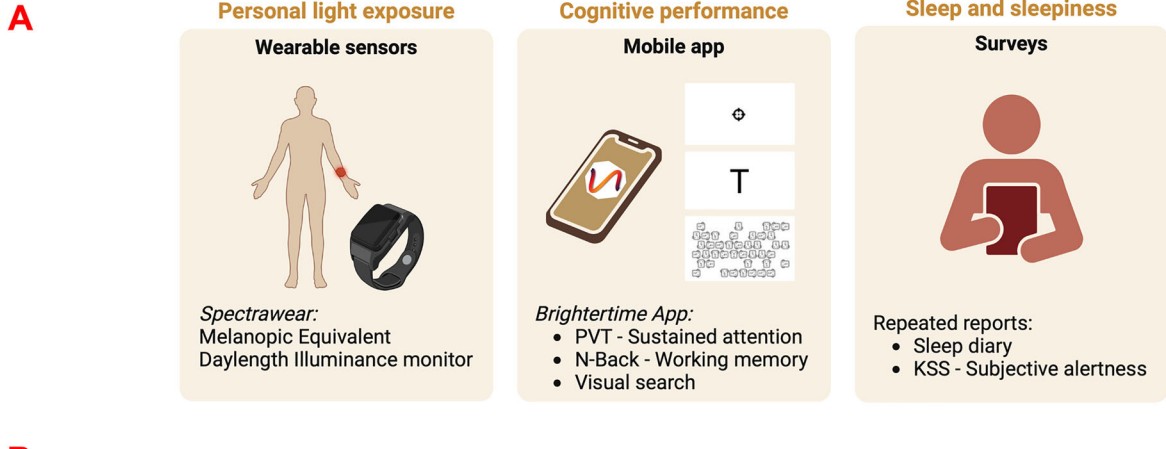

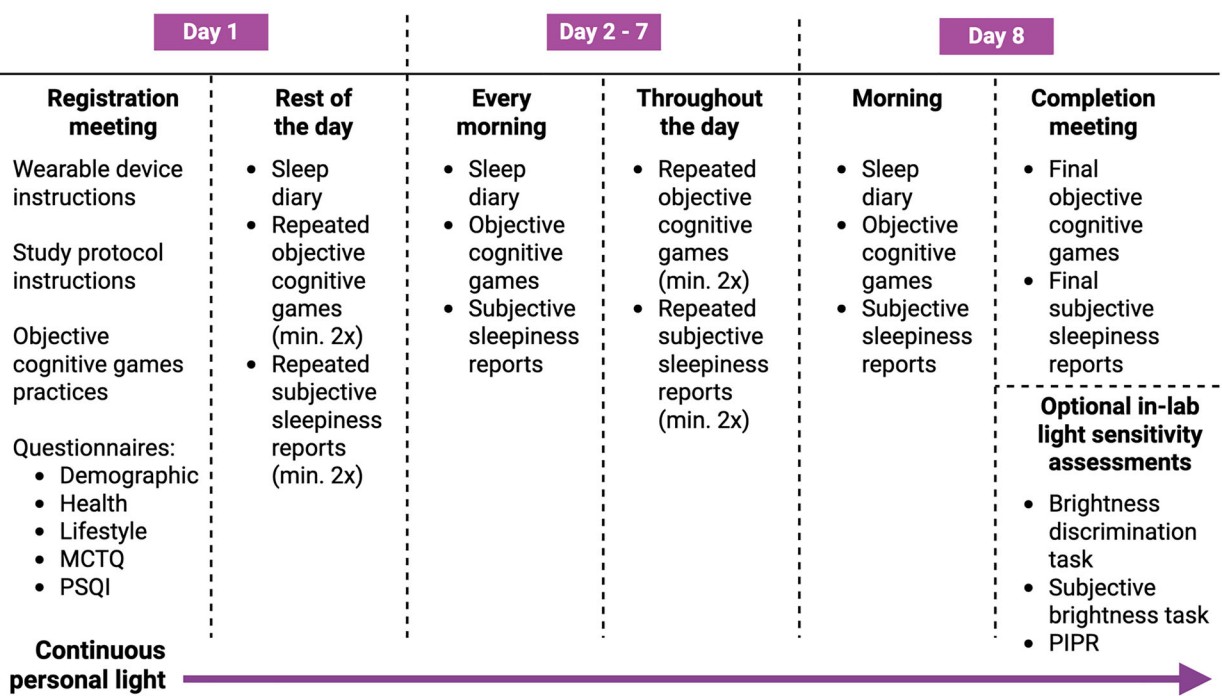

**Fig. 1 | Study protocol. A** The real-world study setting on light and cognition includes three main components: personal light exposure monitoring using the Spectrawear melanopic equivalent daylight illuminance (EDI) sensor, cognitive performance assessment via the Brightertime mobile app (conducting Psychomotor Vigilance Task (PVT), N-back, and Visual Search tasks), and repeated survey collections on sleep and subjective sleepiness. **B** The study lasted for eight days. On the first day, a registration meeting was held, and study instructions were provided. Throughout the following week, participants completed a morning diary, cognitive tasks, and surveys at multiple time points during the day. On the final day, a completion meeting was conducted, and optional visual light sensitivity tasks were performed. Abbreviations: MCTQ Munich Chronotype Questionnaire, PSQI Pittsburgh Sleep Quality Index, PIPR Post-Illumination Pupil Response.

$p < 0.001$, partial $\eta^2 = 0.52$; Time awake$^2$ Std. Coef. = 1.36, 95% CI [1.07, 1.65], $p < 0.001$, partial $\eta^2 = 0.64$) (Fig. 2C). Given the strong relationship between sleep and clock time, it was not surprising to see a similar pattern for subjective sleepiness scores when plotted as a function of time of day, with a mean time of nadir in sleepiness at 14:21 (Sine (time of day) Std. Coef. = 0.24, 95% CI [0.17, 0.30], $p < 0.001$, partial $\eta^2 = 0.48$; Cosine (time of day) Std. Coef. = 0.44, 95% CI [0.36, 0.52], $p < 0.001$, partial $\eta^2 = 0.67$) (Fig. 2D). In comparison, the duration of the previous sleep episode had a relatively smaller effect on subjective sleepiness, with each additional hour of sleep reducing the daily mean subjective sleepiness score by 0.15 points (Std. Coef. = –0.10, 95% CI [–0.15, –0.04], $p = 0.003$, partial $\eta^2 = 0.31$).

Of the cognitive task parameters, median reaction time in the vigilance task showed statistically significant associations with the time of day and

sleep duration, and the false negative ratio in the visual search task showed statistically significant associations with the sleep midpoint (Fig. 2B, Table S3). Thus, the amplitude of time-of-day variation was just 8 ms (Cosine (time of day) Std. Coef. = 0.07, 95% CI [0.02, 0.12], $p = 0.005$, partial $\eta^2 = 0.14$) (Fig. 2E). Each additional hour of sleep reduced the reaction time by 5 ms (Std. Coef. = –0.09, 95% CI [–0.13, –0.04], $p < 0.001$, partial $\eta^2 = 0.08$) (Fig. 2F). Visual search false negative rate decreased by 1% for each additional hour of later sleep midpoint time (Std. Coef. = –0.10, 95% CI [–0.17, –0.03], $p = 0.006$, partial $\eta^2 = 0.007$). Given that vigilance task reaction times are measured on a 100–1000 millisecond scale and visual search accuracy is assessed as a percentage, we acknowledge that, while statistically significant, these changes are relatively subtle in the context of within- and between-participant variability in these parameters.

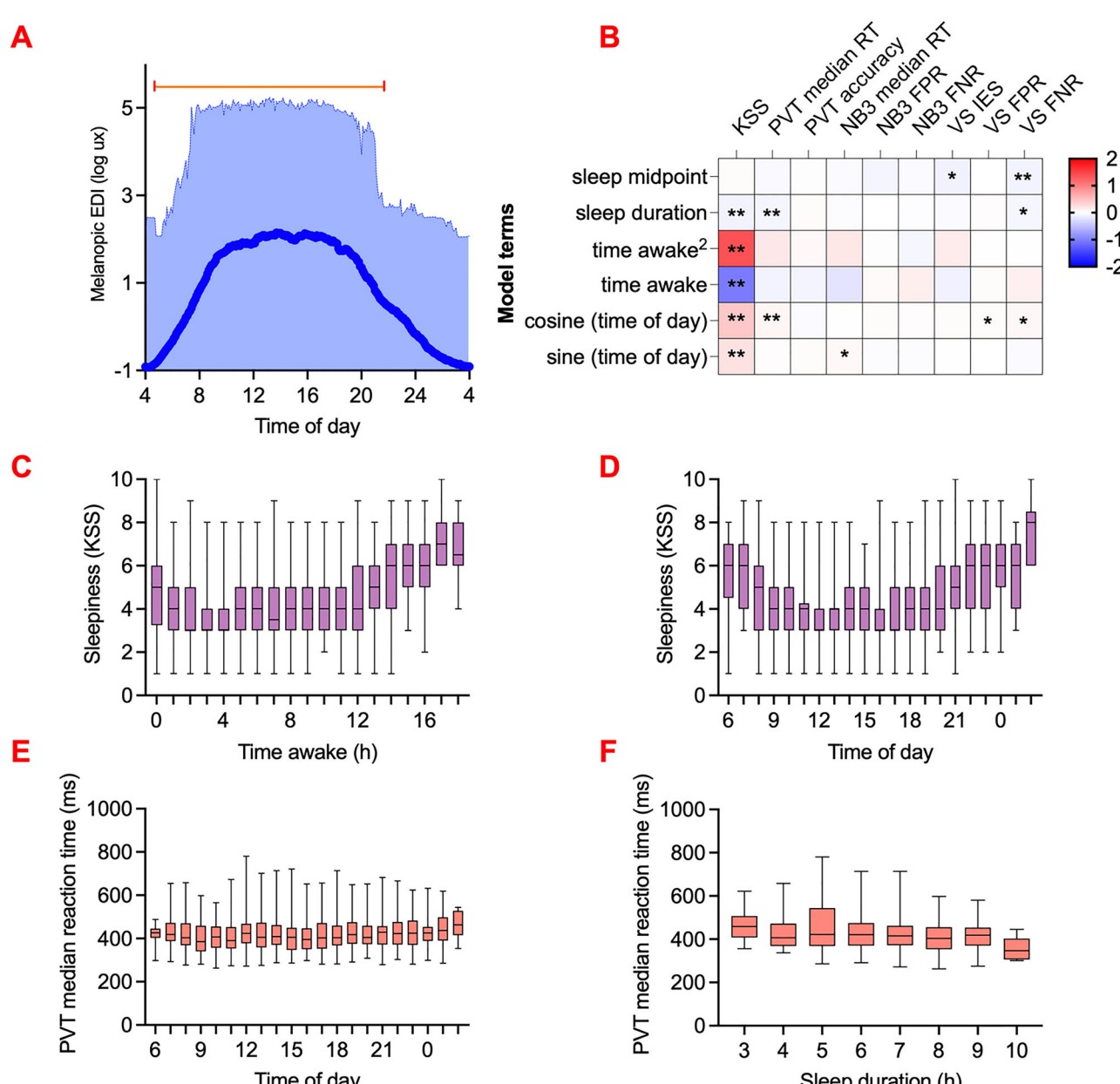

**Fig. 2 | Rhythm of cognitive performance in everyday life. A** Wrist-measured light exposure (mean and range log(melanopic EDI(lux))) across times of day for all observations collected by Spectrawear. The top line indicates daytime, with median sunset and sunrise times marked. **B** Results of the linear mixed model comparing sleep and diurnal rhythm variables to cognitive outcomes (KSS: $n = 1405$, PVT: $n = 1312$, NB3: $n = 1323$, VS: $n = 1319$). Time awake was modelled as a second-degree polynomial function, and time of day was represented as a harmonic fit line using cosine and sine terms. Scale bar indicates standardized coefficients of the linear mixed models. *$p < 0.05$; **$p < 0.008$. KSS Karolinska Sleepiness Scale; PVT Psychomotor Vigilance Task; NB3 3-Back Working Memory Test; VS Visual Search Task; RT reaction time; FPR false positive rate; FNR false negative rate. **C** Change in KSS over time awake. **D** Change in KSS over time of day. **E** Change in PVT median reaction time (ms) over time of day. **F** Change in PVT median reaction time (ms) over sleep duration. **C–F** The box extends from the 25th to the 75th percentile, with the median line inside, and the whiskers extending from the minimum to the maximum.

Subjective sleepiness and cognitive performance were compared between weekdays and weekends. No statistically significant differences were observed for any variable except the visual search IES, which was lower on weekends by 150 ms (95% CI [46.1, 276.7], t(683.6) = 2.7, $p = 0.006$, Cohen's d = 0.16). Daylength during the study ranged from 12.9 to 17.1 h. There was no statistically significant associations between daylength and subjective sleepiness (r(1403) = –0.01, 95% CI [–0.06, 0.04], $p = 0.69$) or vigilance task median reaction times (r(1310) = –0.03, 95% CI [–0.08, 0.03], $p = 0.36$), but small correlations (r < 0.20) were observed between daylength other cognitive measures. The direction of effect

suggested that longer daylengths were associated with reduced cognitive speed and accuracy.

**Cognitive performance is correlated with recent light exposure**
We next set out to ask whether cognitive task performance was correlated with recent light exposure. Including adjustments for time of day, time awake, last sleep duration, and last sleep midpoint, we assessed correlations with current light intensity as the mean light exposure over the preceding 30, 60, 90, and 120 min. This revealed correlations for subjective sleepiness, vigilance task and working memory reaction times, and working memory

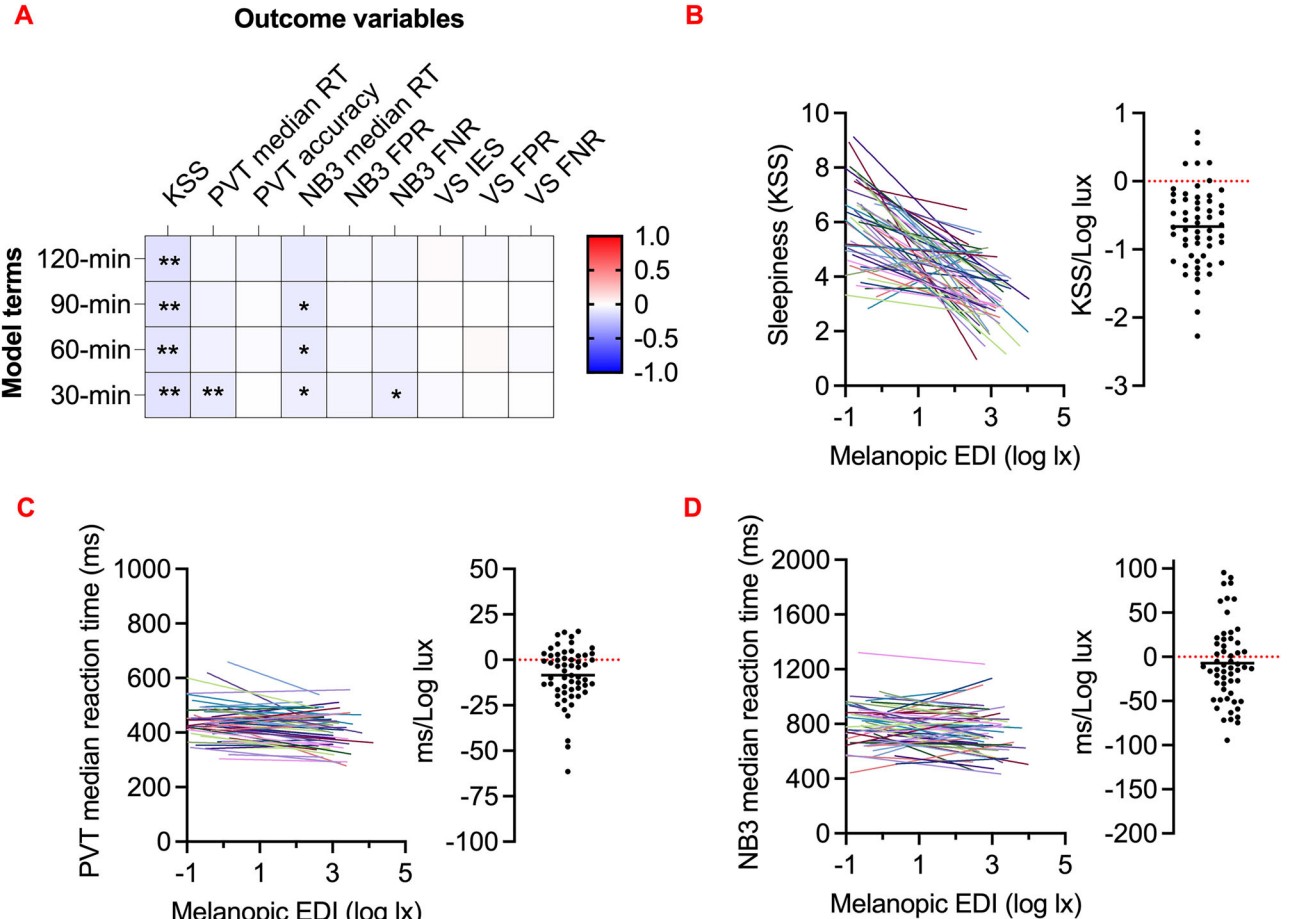

**Fig. 3 | Relationships between recent light exposure and cognitive performance.**
**A** Heatmap shows standardized coefficients from linear mixed models comparing 30-, 60-, 90-, and 120 min light history with cognitive outcomes, including adjustments for time of day, time awake, last sleep duration, and last sleep midpoint. *$p < 0.05$; **$p < 0.013$. KSS Karolinska Sleepiness Scale; PVT Psychomotor Vigilance Task; NB3 3-Back Working Memory Test; VS Visual Search Task; RT reaction time; FPR false positive rate; FNR false negative rate. **B–D** Changes in cognitive performance with melanopic equivalent daylight illuminance (EDI, log lux) history

(KSS: $n = 1234$, PVT: $n = 1222$, NB3: $n = 1212$). **B** Sleepiness (KSS) vs. 30-minute light history, **C** Psychomotor Vigilance Task (PVT) median reaction time (ms) vs. 30 min light history, and **D** 3-Back Working Memory Task (NB3) median reaction time (ms) vs. 30-minute light history. **B–D** The left plots illustrate linear regression lines for each individual. The right distribution plots display the mean and scatter of slopes derived from linear fits for each individual, with NB3 slopes adjusted for the learning effect.

false negative rate. The effect was robust for the 30 min window for vigilance task reaction time and up to the 2 h window for subjective sleepiness, while working memory reaction time showed a moderate but consistent effect up to the 1.5 h window (Fig. 3A, Table S4).

Based on these findings, we utilized 30 min light history durations to model intensity-response relationships. By calculating the slopes of these curves, we quantified the relationship between light exposure and the three cognitive outcomes with most convincing light associations (subjective sleepiness, vigilance task reaction time, and working memory reaction time), offering an indication of cognitive sensitivity to light in real-world environments. On average, a 1 log-lux increase in melanopic EDI was associated with a 0.2-point reduction in subjective sleepiness as measured by the subjective sleepiness (Std. Coef. = −0.11, 95% CI [−0.18, −0.04], $p = 0.003$, partial $\eta^2 = 0.14$) (Fig. 3B). Despite this overall trend, substantial inter-individual variability was observed (including six participants exhibiting positive slopes), indicating that the relationship between light and sleepiness varied among individuals. For the vigilance task, a 4 log-lux increase in melanopic EDI—from the sensor's detection threshold to full sunlight — corresponded to an approximately 30 ms improvement in reaction time (Std. Coef. = −0.09, 95% CI [−0.14, −0.03], $p = 0.003$, partial $\eta^2 = 0.09$) (Fig. 3C). Similarly, but with a small effect size, working memory reaction time for correct short-term memory recall improved by roughly 60 ms

across the same range of melanopic EDI (Std. Coef. = −0.07, 95% CI [−0.13, −0.01], $p = 0.033$, partial $\eta^2 = 0.01$; with task day adjusted for the learning effect) (Fig. 3D). Interestingly, interindividual variability appeared more pronounced for vigilance and working memory performance compared to subjective sleepiness scores, with nearly twice as many participants exhibiting positive slopes in these cognitive tasks. Coefficients of variation were calculated as −88.4 for subjective sleepiness, -181.1 for vigilance, and −633.1 for working memory, highlighting the diversity in correlation with light exposure across different cognitive measures.

**Determinants of light and cognition correlation**
Having assessed within individual relationships between light exposure and performance, we moved to exploring the origins of inter-individual variation in the nature of this association. We first asked whether participants' patterns of light exposure across the week were predictive of their sensitivity. To this end, we extracted various dimensions of light history known to affect circadian rhythms, including daytime and nighttime light intensity, duration of exposure, and the timing and stability of light exposure (see "Methods" for details). These habitual light exposure variables were subsequently compared to the slopes of the relationship between light and cognitive outcomes that demonstrated associations with light history, including subjective sleepiness, vigilance task reaction time, and working

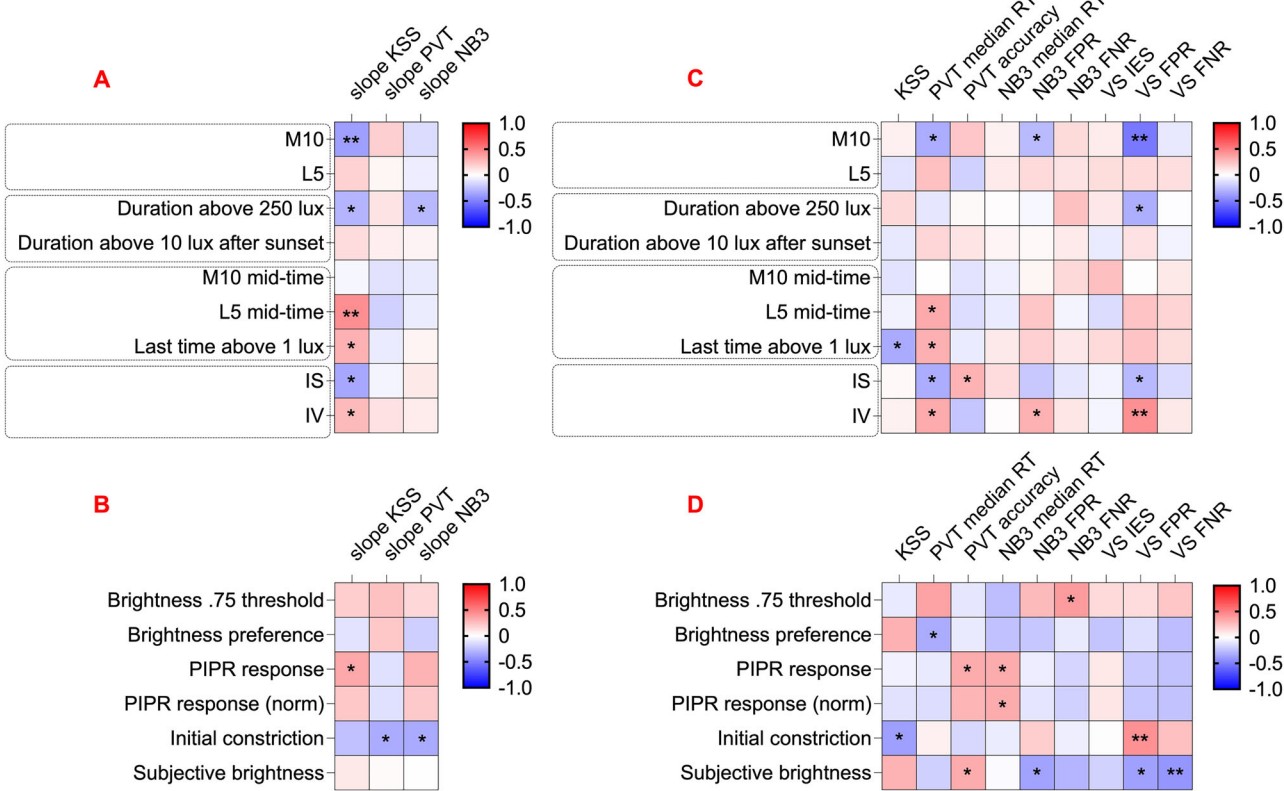

**Fig. 4 | Associations of weekly light exposure variables and laboratory-based photosensitivity measures with real-world photosensitivity of cognitive functions and weekly cognitive performance outcomes.** The comparison of weekly light exposure variables (**A**, **C**) and in-lab photosensitivity measures (**B**, **D**) with real-world photosensitivity of cognitive measures (**A**, **B**) and weekly cognitive outputs (**C**, **D**) is presented. In (**A–D**), heatmaps show Pearson correlation coefficients, with *$p < 0.05$ and (**A**, **C**) **$p < 0.006$, (**B**, **D**) **$p < 0.008$. Sample sizes: **A** KSS: $n = 54$, PVT: $n = 55$, NB3: $n = 54$; **B** KSS: $n = 39$, PVT: $n = 40$, NB3: $n = 39$; **C** $n = 56$ for all models; **D** $n = 41$ for all models. KSS Karolinska Sleepiness Scale; PVT Psychomotor Vigilance Task; NB3 3-Back Working Memory Test; VS Visual Search Task; RT reaction time; FPR false positive rate; FNR false negative rate. Lower slope values indicate higher light sensitivity of KSS, PVT, or NB3, reflecting greater photosensitivity. In (**A**, **C**), light exposure predictors include intensity

variables such as M10 (mean of the brightest 10 h period) and L5 (mean of the dimmest 5-hour period); duration variables such as the time spent above 250 Melanopic EDI lux (minutes) and the time spent above 10 Melanopic EDI lux after sunset (minutes); timing variables including the midpoint time of the M10 period, the midpoint time of the L5 period, and the last time above 1 Melanopic EDI lux; and stability variables such as IS (interdaily stability of light exposure) and IV (intradaily variability of light exposure). In (**B**, **D**), photosensitivity predictors include subjective brightness ratings (0–100), initial pupil constriction (%) in the post-illumination pupil response (PIPR) test, PIPR response (area under the curve between blue and red stimuli up to 6 s after the stimulus), PIPR response normalized to initial constriction, brightness preference (proportion of melanopsin-high choices at the highest melanopsin contrast), and brightness 0.75 threshold (the contrast at which melanopsin-high was chosen 75% of the time).

memory reaction time (Table S5). Higher M10, earlier clock times for the dimmest 5 h epoch and light exposure patterns with lower intra-daily variability and higher inter-daily stability all correlated with stronger associations between recent light exposure and subjective sleepiness. The strongest correlations were observed in the relationship between light and subjective sleepiness and timing of the main epoch of darkness (indicative of time in bed) (Fig. 4A). Specifically, participants whose dark epoch occurred later and had lower daytime light exposure showed less steep negative slopes for subjective sleepiness, indicating a weaker association between light and subjective sleepiness (L5 mid-time: $r(52) = 0.44$, $p < 0.001$, 95% CI [0.20, 0.64] and M10: $r(52) = -0.38$, $p = 0.005$, 95% CI [−0.59, −0.12]; Figure S5A). These correlations remained robust after adjusting for age, sex, caffeine consumption, alcohol consumption, smoking, chronotype (MSFsc), and sleep problems (PSQI) ($b = 0.22$, SE = 0.08, 95% CI [0.05, 0.38], $p = 0.013$ and $b = -0.76$, SE = 0.29, 95% CI [−1.35, −0.17], $p = 0.013$ respectively).

We next explored whether it was possible to predict cognitive task photosensitivity using in-lab assessments of light sensitivity. To this end, a subgroup of participants ($n = 41$) volunteered for in-lab assessments designed to measure melanopsin sensitivity through a series of pupillometric and perceptual psychophysics tasks[43]. The lab-based tests included the Post-Illumination Pupil Response (PIPR) test, silent substitution melanopic brightness discrimination tasks, and a subjective brightness

assessment. The derived variables from these tasks were then analysed in relation to the real-world light sensitivity of cognitive functions mentioned earlier (Table S6). This failed to reveal strong associations (Fig. 4B).

## Determinants of inter-individual differences in cognitive performance

We next asked whether any aspects of an individual's light exposure profile correlated with overall cognitive performance (Table S7). Once again, we used nine cognitive performance measures (subjective sleepiness, vigilance task, working memory, and visual search), which were averaged across the week. The most consistent associations across endpoints were with the M10 (intensity over the day's brightest 10 h) and IV (intra-daily variability) (Fig. 4C). Specifically, people with brighter daytime exposure (M10; Fig. S5B) and less fragmented daily patterns of light exposure (IV) had lower visual search false positive rate ($r(54) = -0.53$, $p < 0.001$, 95% CI [−0.69, −0.31] and $r(54) = 0.44$, $p < 0.001$, 95% CI [0.20, 0.63] respectively) and lower working memory false positive rate ($r(54) = -0.27$, $p = 0.043$, 95% CI [−0.50, −0.01] and $r(54) = 0.31$, $p = 0.021$, 95% CI [0.05, 0.53] respectively), and reduced vigilance task reaction time ($r(54) = -0.32$, $p = 0.015$, 95% CI [−0.54, −0.07] and $r(54) = 0.33$, $p = 0.013$, 95% CI [0.08, 0.55] respectively). Correlations of visual search false positive rate with M10 and IV remained robust after adjusting for covariates ($b = -14.4$,

SE = 3.30, 95% CI [–21.0, –7.7], $p < 0.001$ and $b = 37.7$, SE = 12.7, 95% CI 12.1, 63.4], $p = 0.005$ respectively).

Lastly, in-lab light sensitivity measures were compared to cognitive variables (Table S8). Initial pupil constriction had the most significant associations, with greater constriction indicative of greater false positives in visual search ($r(39) = 0.43$, $p = 0.005$, 95% CI [0.15, 0.65]; Fig. 4D). In addition, a higher rating of subjective brightness for a standard test stimulus was correlated with fewer false negatives in visual search ($r(39) = –0.42$, $p = 0.007$, 95% CI [–0.64, –0.12]; Fig. S5C). These associations remained robust after adjusting for demographic and lifestyle covariates ($b = 0.70$, SE = 0.19, 95% CI [0.32, 1.09], $p = 0.001$ and $b = –0.24$, SE = 0.10, 95% CI [–0.44, –0.04], $p = 0.021$ respectively).

## Discussion

This study examined the relationship between light exposure and cognitive performance in real-world settings. Our findings are consistent with the hypothesis that even outside controlled laboratory conditions, where participants continued their daily routines, both recent and long-term light exposure positively influences cognitive performance. We find that recent bright light correlates with lower sleepiness scores and faster reaction times for vigilance task and working memory tasks. Meanwhile, participants whose habitual light exposure was characterized by brighter light through daytimes and less intra-daily fragmentation had fewer false negative responses in working memory and visual search and reduced reaction time in the vigilance task. Our study was less successful in identifying predictors of inter-individual differences in the strength of these light associations. Earlier bedtimes (as estimated by the time of dimmest light) and brighter daytime exposure were associated with stronger light dependence for subjective sleepiness, but this was not strongly apparent for other endpoints. Similarly, although there were some statistically significant associations between the strength of light associations and performance in a bank of putative in-lab assessments of melanopic responsiveness these did not form a clear pattern.

The strongest intra-individual associations with light were observed for subjective sleepiness, with nearly all participants experiencing reduced sleepiness following bright light exposure independent of time of day. This is consistent with a field study demonstrating that light exposure reduces momentary exhaustion, as well as our previous findings indicating that light exposure decreases subjective sleepiness[41,47]. This relationship was strongest in individuals with earlier bedtimes (inferred from the timing of lowest light exposure L5). Given that there was no statistically significant association between the timing of L5 and average subjective sleepiness this implies that, compared with their peers with later bedtimes, those with earlier bedtimes tend to be both more reliably wakeful under bright- and sleepy under dim-light. This could enforce an earlier sleep phase because morning light tends to be bright and evening light dim.

Surprisingly, neither time of day nor time awake had large effects on vigilance task, visual search, or working memory performance. Given that attentional and memory-related brain regions exhibit intrinsic circadian rhythms, and that sleep prior to cognitive tasks is a crucial determinant of cognitive functioning, one might expect a diurnal cycle in these cognitive traits[1,48]. However, its effects were not clearly observable in our data. The effect of light was found to be stronger than the effect of time of day. This remains an ongoing area of research in the literature, with inconsistent findings. Some studies suggest that circadian effects on executive function are limited compared to arousal-related measures[1,27]. Additionally, external factors in everyday life—such as lifestyle preferences, motivation, and environmental distractions—can create significant intra- and inter-individual differences, potentially overriding internal oscillatory effects[49].

The correlation between higher recent light and reduced reaction times observed across both vigilance task and working memory is consistent with controlled laboratory findings[12,17,19,20,22]. Our results indicate that exposure to bright light, comparable to daylight, may improve reaction times in these tasks by 7–10% compared to dim conditions. While this effect size is modest, they are larger than would be expected for visual adaptation alone[50,51], and

these improvements in cognitive performance may have practical implications for health, safety, and work efficiency, particularly in low-light workplaces, during extended work hours, or night shifts[31–35]. Previous research has shown that adverse circadian and homeostatic processes lead to a global reduction in vigilance but do not necessarily impair visual selective attention[52]. Notably, the improvements in reaction time were not accompanied by changes in accuracy, indicating that this was not merely a general hyper-responsiveness—participants continued to make correct responses despite responding more quickly. Other aspects of task performance appeared less dependent on light exposure, suggesting either that light is not a major determinant of working memory and visual search efficiency or that our study lacked the statistical power to detect such effects.

Our analysis of light exposure history further revealed that cognitive outcomes are correlated with habitual light exposure patterns. Individuals whose light exposure patterns aligned more closely with recommended guidelines[9,53] – including brighter daytime exposure, an earlier sleep phase, and more stable and consistent daily light exposure – tended to perform better on cognitive tasks. However, these effects were not uniformly observed across all cognitive parameters or all exposure-performance correlations. The strongest associations emerged with the visual search task, suggesting that prolonged exposure to an optimal light environment may particularly enhance visual attention and processing efficiency.

Overall, two key patterns emerged from our results: (1) recent light exposure was associated with cognitive effects consistent with increased arousal (e.g., heightened alertness and faster reaction times), and (2) a history of bright, stable daytime light exposure was linked to enhanced sustained attention in a visual search task. Both effects are likely initiated by activation of the ipRGC system, as laboratory studies have demonstrated a short-wavelength bias in the acute effects of light on arousal and cognitive function (consistent with melanopsin spectral sensitivity)[7,13,22], alongside the established role of ipRGCs in regulating central clock function via the retinohypothalamic tract to the suprachiasmatic nucleus (SCN)[54]. Acute effects may involve not only ipRGC input to the SCN but also contributions from projections to other brain regions. For example, the locus coeruleus (LC) and its ascending reticular activating system have been implicated in mediating acute arousal, potentially through indirect input from the SCN[55,56]. However, other ipRGC target regions may also contribute. Animal studies have identified several candidate pathways, though their relevance to humans remains less certain[57]. Furthermore, improved attention under high-melanopic light suggests contributions of top-down attentional and executive mechanisms, likely mediated by ipRGC–thalamic–corticolimbic pathways influencing prefrontal cortical activity[4,5,58]. For the longer-term effects, exposure to high-amplitude, early light patterns appears to strengthen both circadian robustness and sleep homeostasis, thereby supporting improved cognitive performance over time[23,59].

Previous studies on circadian rhythms have identified substantial interindividual differences in light sensitivity, particularly concerning melatonin suppression[60]. It would be ideal to have an accessible biomarker for light sensitivity. However, none of the battery of in-lab tests we conducted robustly predicted light sensitivity across cognitive tasks. While some statistically significant associations were observed, their reliability was limited. This suggests that the in-lab measures used here do not function as global predictors of photosensitivity. Perhaps in-lab and in-field measures reflect different processes with independent regulators of sensitivity. Alternatively, sensitivity could change so dynamically that a one-off in-lab measure is a poor predictor of sensitivity across other conditions; or our sample may simply not have captured a sufficiently broad range of individual sensitivities to detect a meaningful association.

Perhaps surprisingly, our data did reveal relationships between average task performance per individual and their in-lab light responsiveness. The magnitude of initial pupil constriction and the subjective brightness score for a standard stimulus each correlated with aspects of mean task performance, albeit without showing consistent effects across tasks. Greater pupil reactivity was associated with reduced visual search accuracy, while greater subjective brightness score was linked to higher visual search accuracy. The

basis for these associations remains uncertain. In principle, people with higher light sensitivity may have improved cognitive performance across lighting conditions. However, the value of these in lab assays as predictors of melanopic light sensitivity under different circumstances is unproven and if this were the origin for these correlations, we might expect them to be strongest for cognitive parameters with the clearest associations with recent light (subjective sleepiness and reaction times for vigilance task and working memory). The pupil can provide an indication of central arousal mechanisms and has been shown to correlate with vigilance performance[61], but it would be surprising if a one-off in lab measure predicted average arousal over the previous 7 days.

## Limitations

This study has several limitations. Our sample primarily consisted of individuals without significant circadian or light exposure challenges, leaving the possibility that the observed effects could be more (or less) pronounced in populations with disrupted circadian rhythms, such as shift workers, individuals with sleep and psychiatric disorders, or older adults. Additionally, because of small sample size, variations in photosensitivity likely influenced[62] by factors such as age and genetics, were not explicitly controlled for in this study. Importantly, as a correlational study rather than an intervention, our findings cannot establish causality between light exposure and cognitive performance. However, our protocol successfully demonstrated the feasibility of monitoring personal light exposure and cognitive performance in real-world settings and several of our findings are consistent with the hypothesis that light can be a determinant of cognitive function in everyday life.

## Data availability

The full study protocol was shared in protocols.io (https://doi.org/10.17504/protocols.io.n92ldrjjxg5b/v1). Anonymized data on light exposure, in-lab light sensitivity assessments, and cognitive tasks, as well as the processed data used to generate the figures and tables in the paper, are available in a Figshare repository (https://doi.org/10.48420/28911977).

## Code availability

R code to process and analyze the data of light exposure and cognitive tasks created for the study are available in a GitHub repository (https://github.com/altugdidikoglu/light-cognition-inreallife). Software and hardware designs of the wearable light dosimeter are available in a repository (https://github.com/Non-Invasive-Bioelectronics-Lab/Wearable_Light_Sensor_Public).

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

## Acknowledgements

This work was funded by the University of Manchester Wellcome Trust/ISSF fund to R.J.L., M.V.T., T.M.B., A.J.C., and S.J. and by a Wellcome Trust Investigator Award (210684/Z/18/Z) to R.J.L. The funders had no role in study design, data collection and analysis, decision to publish or preparation of the manuscript.

## Author contributions

A.D.: Project administration, methodology, recruitment, data collection, data analysis; T.W.: Methodology, data analysis; L.B.: Methodology, recruitment, data collection; N.M., A.J.C., T.M.B. and A.D.: Design, production, calibration, technical support of the wearable light dosimeters; M.V.T., T.M.B., S.J., A.J.C. and R.J.L.: Conceptualization, supervision, methodology. A.D., T.W. and R.J.L.: Writing-original draft. All authors discussed the results and edited the manuscript. All authors have read and agreed to the published version of the manuscript.

## Competing interests

R.J.L. and T.M.B. have received investigator-initiated grant funding from Signify/Philips Lighting and R.J.L. has received honoraria from Samsung Electronics. All other authors declare no competing interests.
