## [Transparent Peer Review file · Communications Psychology]

Relationships between light exposure and aspects of cognitive function in everyday life

Corresponding Author: Dr Altug Didikoglu

Version 0:

Decision Letter:

Dear Dr Didikoglu,

Thank you for submitting your manuscript titled "Relationships between light exposure and aspects of cognitive function in a group of UK adults in everyday life" to Communications Psychology. We have given the paper our careful consideration and find it of potential interest. However, due to certain shortcomings we are concerned that sending the current manuscript out to review could lead to unnecessary delays and quite possibly an undesirable outcome of the review process.

In particular, we require full reporting of statistics for all analyses. Indicating effects and suggesting significance visually (e.g., asterisks in the heatmaps) is not sufficient. Frequentist inferential statistics should be reported as follows: statistics (degrees of freedom) = value, p = value, effect size statistics = value, % Confidence Intervals = values. More detailed information on this requirement is included in the attached checklist and on our website (<https://www.nature.com/commspsychol/submit/submission-guidelines#statistical-guidelines>).

We would therefore like to invite you to revise your manuscript to address these concerns before we make a final determination on whether to send your manuscript for external review.

We shall hope to receive your revised version as soon as you are able to complete the suggested revisions. If something similar is published in the interim we will have to consider the impact it has on the novelty of a revised manuscript.

If you anticipate a delay of more than four weeks, please let us know. Should your manuscript be substantially delayed without notifying us in advance and your article is eventually published, the received date may be that of the revised, not the original, version.

We also ask that you ensure your manuscript complies with our editorial policies and reporting requirements.

To that end, we require revised manuscripts to be accompanied by two completed items: a reporting summary that collects information on study design and procedure, and an editorial policy checklist that verifies compliance with all required editorial policies.

- <https://www.nature.com/documents/nr-reporting-summary.zip>>Nature Research Reporting Summary
- <https://www.nature.com/documents/nr-editorial-policy-checklist.pdf>>Editorial Policy Checklist

All points on the policy checklist must be addressed. Your revised manuscript can only be sent to referees if these checklists are completed and uploaded with the revision.

If you are not interested in submitting a suitably revised manuscript in the future please let me know immediately so we can close your file. If you have any questions, please contact me.

Please use the link below when you are prepared to resubmit.
Link Redacted

Thank you for your interest in Communications Psychology.

Best regards,
Xiaoqing Hu

Xiaoqing Hu, PhD
Editorial Board Member
Communications Psychology
orcid.org/0000-0001-8112-9700

Version 1:

Decision Letter:

Dear Dr Didikoglu,

Thank you for your patience during the peer-review process. Your manuscript titled "Relationships between light exposure and aspects of cognitive function in a group of UK adults in everyday life" has now been seen by 2 reviewers, and I include their comments at the end of this message. They find your work of interest but raised some important points. We are interested in the possibility of publishing your study in Communications Psychology, but would like to consider your responses to these concerns and assess a revised manuscript before we make a final decision on publication.

We therefore invite you to revise and resubmit your manuscript, along with a point-by-point response to the reviewers. Please highlight all changes in the manuscript text file.

Editorially, we consider it crucial that the reviewers' methodological concerns, such as the potential confounding influence of seasonal variations in light exposure, are thoroughly addressed in the revised manuscript.

I am attaching an Editorial Requests Table that details critical reporting requirements for the revised manuscript. Please attend to each item and ensure your manuscript is fully compliant. If your revised manuscript is not aligned with these requests on major issues, such as those concerning statistics, it may be returned to you for further revisions without re-review.

Please submit the following items:

- Revised manuscript
- Point-by-point response to the referees' comments
- Cover letter (as a separate document)
- <https://www.nature.com/documents/nr-reporting-summary.pdf> Nature Research Reporting Summary
- Completed Editorial Request Table (attached).

via this link: Link Redacted .

Additional guidance is available in our style and formatting guide <https://www.nature.com/documents/commpsychol-style-formatting-guide-accept.pdf> Communications Psychology formatting guide

We hope to receive your revised paper within 8 weeks; please let us know if you aren't able to submit it within this time so

that we can discuss how best to proceed. If we don't hear from you, and the revision process takes significantly longer, we may close your file. In this event, we will still be happy to reconsider your paper at a later date, provided it still presents a significant contribution to the literature at that stage.

Best regards,

Troby Lui, on behalf of

Xiaoqing Hu

Troby Lui, PhD
Associate Editor
Communications Psychology

Xiaoqing Hu, PhD
Editorial Board Member
Communications Psychology
orcid.org/0000-0001-8112-9700

REVIEWER EXPERTISE:

Reviewer #1: sleep

Reviewer #2: sleep, daytime vigilance, cognition

REVIEWER REPORTS:

Reviewer #1 (Remarks to the Author):

Thank you for the opportunity to review this study, which explores the relationship between light exposure and cognitive performance in a real-world setting. I have a few suggestions for your consideration:

1. Could you provide more details on how compliance with the cognitive tasks in the mobile app is measured, especially since these tasks are not conducted in a laboratory setting? This information would enhance the reliability of the findings. Also are these cognitive tasks validated?
2. How do you account for potential practice effects in the cognitive tasks? Addressing this could strengthen the study's conclusions regarding cognitive performance.
3. It would be helpful to include information about the sensitivity and validation of the light logger used in the Spectrawear device. Understanding its accuracy is crucial for interpreting the data.
4. Assessment of Sleep Disorders: How were sleep disorders assessed in the participants? Additionally, considering other psychiatric conditions that could influence cognitive performance would be important for a comprehensive analysis.
5. Did you analyze any potential gender differences in the associations between light exposure and cognitive performance? This could provide valuable insights into the generalizability of your findings.
6. Subgroup Analysis: For the six participants who exhibited positive slopes between light exposure and sleepiness, are there specific characteristics within this subgroup that might explain these trends? Such as chronotype?
7. Did you observe any differences in cognitive performance and light exposure between weekdays and weekends? This could reveal important patterns related to daily routines.
8. Are there associations between total daytime light exposure, sleep patterns, and cognitive performance? Investigating this relationship could add depth to your findings.
9. The discussion section could benefit from a deeper exploration of the mechanisms linking light exposure to cognitive performance, rather than solely describing the findings. This would help contextualize the implications of your research.

Reviewer #2 (Remarks to the Author):

This is an interesting study that provides valuable insights into the relationship between light exposure and cognitive function in real-world settings. The results are clear and the conclusions are suitable for them. I only have a few questions.

- (1) Given the study's temporal scope spanning July 2022 to August 2023, have seasonal variations in ambient light exposure been systematically evaluated as a potential confounding factor? The observed cross-seasonal differences in daylight duration and intensity could theoretically contribute to inter-individual variability in cognitive outcomes.
- (2) There may be differences in the diurnal patterns of cognitive function between habitual midday nappers and non-habitual nappers. Has the study examined the potential moderating role of midday napping habits on the diurnal variation of cognitive performance?

- (3) There is no mention of correction for multiple comparisons for the correlation analyses, even though a large number of bivariate relationships were tested (i.e., ≥ 3 correlations between light exposure and performance).
- (4) Please add some discussions on the neural mechanisms in the Discussion section.
- (5) Some of the references were outdated.

Version 2:

Decision Letter:

Dear Dr Didikoglu,

Your manuscript titled "Relationships between light exposure and aspects of cognitive function in a group of UK adults in everyday life" has now been seen by our reviewers, whose comments appear below. In light of their advice I am delighted to say that we are happy, in principle, to publish a suitably revised version in Communications Psychology.

We therefore invite you to revise your paper one last time to address the remaining concerns of our reviewers and a list of editorial requests. At the same time we ask that you edit your manuscript to comply with our format requirements and to maximise the accessibility and therefore the impact of your work.

EDITORIAL REQUESTS:

SUBMISSION INFORMATION:

OPEN ACCESS:

At acceptance, you will be provided with instructions for completing the open access licence agreement on behalf of all authors. This grants us the necessary permissions to publish your paper. Additionally, you will be asked to declare that all required third party permissions have been obtained, and to provide billing information in order to pay the article-processing

charge (APC).

* **DATA AVAILABILITY:**

Link Redacted

Best regards,

Troy Lui, on behalf of

Xiaoqing Hu

Troy Lui, PhD
Associate Editor
Communications Psychology

Xiaoqing Hu, PhD
Editorial Board Member
Communications Psychology
orcid.org/0000-0001-8112-9700

REVIEWERS' COMMENTS:

Reviewer #1 (Remarks to the Author):

Thank you for addressing all my comments

Reviewer #2 (Remarks to the Author):

This manuscript has undergone a thorough revision based on the initial review comments. The authors have addressed all major concerns raised during the first round of review. After careful consideration, I recommend acceptance of this manuscript for publication in Communications Psychology.

REVIEWER REPORTS:

Reviewer #1 (Remarks to the Author):

Thank you for the opportunity to review this study, which explores the relationship between light exposure and cognitive performance in a real-world setting. I have a few suggestions for your consideration:

1. Could you provide more details on how compliance with the cognitive tasks in the mobile app is measured, especially since these tasks are not conducted in a laboratory setting? This information would enhance the reliability of the findings. Also are these cognitive tasks validated?

Answer: Thank you for this valuable comment. This study was designed as a real-world investigation using ecological momentary assessments, with the goal of capturing both within- and between-individual variance in cognitive performance under everyday conditions. In our earlier work (Gardesevic et al., 2022), we developed the BrighterTime mobile application. The psychomotor vigilance task (PVT), N-back, and visual search tasks implemented in the app were designed to match validated laboratory paradigms in terms of number of trials, stimulus characteristics, and interstimulus durations. (page 14 of the manuscript) Subsequently, an optimization phase was conducted to reduce task duration and trial numbers to a minimum, while preserving the mean and variance of the results. That study confirmed that the tasks were both feasible and engaging in real-world contexts, with sufficient validity to capture cognitive fluctuations in daily life. Cognitive scores varied considerably within and across individuals on a day-to-day basis, reflecting both circadian rhythmicity and trait-level differences. Unlike laboratory tasks, real-world conditions introduce distractions, but repeated measurements across time provide a reliable means of capturing trends and rhythmicity.

Regarding compliance, participants were encouraged to provide multiple entries throughout the day (page 14 of the manuscript), with a recommended schedule of morning, midday, and evening reports, but without rigid enforcement. (page 15 of the manuscript) To ensure sufficient data quality for analysis, we applied preprocessing steps: excluding trials with median reaction times < 100 ms, accuracy < 30%, and participants with fewer than 7 entries. Out of 4,049 total raw task submissions (PVT, NB3, and VS), 4,015 (~99%) remained after cleaning, indicating strong compliance and data quality. Furthermore, compliance can be quantified by the number of entries: participants contributed between 13 and 52 entries (median = 23) across the study week, yielding an average of 3.2 entries per day, which aligns well with our recommended protocol (page 3 of the manuscript).

2. How do you account for potential practice effects in the cognitive tasks? Addressing this could strengthen the study's conclusions regarding cognitive performance.

Answer: We appreciate this important observation. Because our design involved repeated cognitive assessments, we considered the potential for practice, learning, and task order effects. To address these, several measures were implemented. First, the app randomized the presentation order of the tasks within each session (page 14 of the manuscript). Second, upon first logging into the app, participants completed a short practice session for each cognitive task to minimize early learning effects (page 14 of the manuscript). As detailed in the manuscript (page 15 and Figure S5), the PVT and the KSS were unaffected by the number of tasks completed per day and did not show evidence of practice effect. In contrast, the NB3 and visual search VS tasks, which are more complex, exhibited mild learning effects: participants became slightly faster and more accurate over time, particularly in the VS inverse efficiency score (IES). To control for this, we included "task day" as a covariate in the statistical models for NB3 and VS where necessary, thereby accounting for potential confounding due to practice effects.

3. It would be helpful to include information about the sensitivity and validation of the light logger used in the Spectrawear device. Understanding its accuracy is crucial for interpreting the data.

Answer: Thank you for this valuable suggestion. We have added further detail to **page 13** of the manuscript. The Spectrawear device is a multichannel light sensor specifically designed to measure melanopic EDI. Above 1 lx, the device demonstrated an average mean absolute log deviation of < 0.05 log units, with a minimum between-device correlation of 0.99. Its typical performance estimates α -opic EDIs within $\pm 15\%$ across a wide dynamic range (1 to 100,000 lx). In addition, the angular response to incident light was with a half-width at half maximum of 51° . These characteristics confirm that the device provides robust and accurate light exposure measurements suitable for the present study.

4. Assessment of Sleep Disorders: How were sleep disorders assessed in the participants? Additionally, considering other psychiatric conditions that could influence cognitive performance would be important for a comprehensive analysis.

Answer: Thank you for raising this important point. Our study was designed to capture cognitive performance in a real-world population, and therefore we did not apply specific health-related exclusion criteria. This approach increases ecological validity but also introduces certain limitations in detecting or disentangling the influence of clinical conditions. We have addressed this in the manuscript (**page 12**), noting that: "Our sample primarily consisted of individuals without significant circadian or light exposure challenges, leaving the possibility that the observed effects could be more (or less) pronounced in populations with disrupted circadian rhythms, such as shift workers, individuals with sleep and psychiatric disorders, or older adults." Sleep and psychiatric conditions were assessed subjectively at baseline through self-report of previous diagnoses. In addition, sleep quality was evaluated with the Pittsburgh Sleep Quality Index (PSQI), which indicated generally good sleep health among participants (mean = 6.4, SD = 1.9; **page 13**). A small subset of participants reported conditions such as ADHD, anxiety, and depression. While we recognize that these disorders and associated medications can affect cognition, the limited sample size and lack of detailed information on disorder subtype, severity, and treatment precluded meaningful statistical comparisons across these groups.

5. Did you analyze any potential gender differences in the associations between light exposure and cognitive performance? This could provide valuable insights into the generalizability of your findings.

Answer: Thank you for this thoughtful suggestion. As described in **Section 2.4** (Comparison of light sensitivity and light exposure) and **Section 2.5** (Comparison of cognitive performance and light exposure), we repeated all significant analyses with sex included as an adjustment factor (methods detailed on **page 17**). The results, reported on **pages 9-10**, indicated that sex had no significant effect on the associations examined. These findings suggest that the reported relationships between light exposure and cognitive performance were consistent across genders in our sample.

6. Subgroup Analysis: For the six participants who exhibited positive slopes between light exposure and sleepiness, are there specific characteristics within this subgroup that might explain these trends? Such as chronotype?

Answer: In **Section 2.5**, we examined light sensitivity slopes in relation to light exposure using multivariate linear regression, adjusting for age, sex, daylength, caffeine consumption, alcohol consumption, smoking, chronotype (MSFsc), and sleep problems (PSQI). None of these covariates significantly influenced the light sensitivity slopes. We also conducted a categorical

comparison between participants with positive slopes and those with negative slopes across the same set of variables (age, sex, daylength, caffeine consumption, alcohol consumption, smoking, chronotype, and sleep problems). Again, none of the collected covariates accounted for the observed interindividual differences in sensitivity.

7. Did you observe any differences in cognitive performance and light exposure between weekdays and weekends? This could reveal important patterns related to daily routines.

Answer: We thank the reviewer for this insightful suggestion. We have added a new analysis comparing cognitive performance and light exposure between weekdays and weekends. The results indicated no significant differences for most variables. On **page 6**, we now state: "Subjective sleepiness and cognitive performance were compared between weekdays and weekends. No significant differences were observed for any variable except the VS IES, which was slightly lower on weekends by 150 ms ($t(683.6) = 2.7$, $p = 0.006$, Cohen's $d = 0.16$)." These findings suggest that, in our sample, cognitive performance does not differ meaningfully between weekdays and weekends, and no further adjustments are necessary.

8. Are there associations between total daytime light exposure, sleep patterns, and cognitive performance? Investigating this relationship could add depth to your findings.

Answer: We have examined associations between sleep patterns and cognitive performance (**page 6** of the manuscript). Specifically, the duration of the previous sleep episode had a modest effect on the KSS, with each additional hour of sleep reducing the daily mean KSS score by 0.15 points. Each additional hour of sleep also reduced reaction time by approximately 5 milliseconds. Additionally, the VS FNR decreased by 1% for each hour of later sleep midpoint. We further analyzed the relationship between recent light exposure and cognitive variables while adjusting for sleep duration and midpoint (Methods, **page 16**). We agree that there may be a three-way interaction between light, sleep, and cognition; however, such analyses are beyond the scope of the present paper and would require more complex modelling. Importantly, our dataset will be made publicly available, enabling future studies to explore these relationships in greater depth.

9. The discussion section could benefit from a deeper exploration of the mechanisms linking light exposure to cognitive performance, rather than solely describing the findings. This would help contextualize the implications of your research.

Answer: We thank the reviewer for this valuable suggestion. In response, we have added a paragraph to the **Discussion** section that summarizes potential neural mechanisms underlying our findings: "Overall, two key patterns emerged from our results: (1) recent light exposure was associated with cognitive effects consistent with increased arousal (e.g., heightened alertness and faster reaction times), and (2) a history of bright, stable daytime light exposure was linked to enhanced sustained attention in a visual search task. Both effects are likely initiated by activation of the ipRGC system, as laboratory studies have demonstrated a short-wavelength bias in the acute effects of light on arousal and cognitive function (consistent with melanopsin spectral sensitivity), alongside the established role of ipRGCs in regulating central clock function via the retinohypothalamic tract to the suprachiasmatic nucleus (SCN). Acute effects may involve not only ipRGC input to the SCN but also contributions from projections to other brain regions. For example, the locus coeruleus (LC) and its ascending reticular activating system have been implicated in mediating acute arousal, potentially through indirect input from the SCN. However, other ipRGC target regions may also contribute. Animal studies have identified several candidate pathways, though their relevance to humans remains less certain. Furthermore, improved attention under high-melanopic light suggests contributions of top-down attentional and executive mechanisms, likely mediated by ipRGC–thalamic–corticolimbic pathways influencing prefrontal cortical activity. For the longer-term effects, exposure to high-amplitude, early light patterns appears to strengthen both circadian

robustness and sleep homeostasis, thereby supporting improved cognitive performance over time.”

Reviewer #2 (Remarks to the Author):

This is an interesting study that provides valuable insights into the relationship between light exposure and cognitive function in real-world settings. The results are clear and the conclusions are suitable for them. I only have a few questions.

1. Given the study's temporal scope spanning July 2022 to August 2023, have seasonal variations in ambient light exposure been systematically evaluated as a potential confounding factor? The observed cross-seasonal differences in daylight duration and intensity could theoretically contribute to inter-individual variability in cognitive outcomes.

Answer: We thank the reviewer for this insightful suggestion. We have conducted a new analysis to assess potential effects of seasonal variation in ambient light exposure on cognitive performance (**page 6** of manuscript). Daylength during the study ranged from 12.9 to 17.1 hours. Increasing daylength was not associated with KSS or PVT median reaction times, though small correlations ($r < 0.20$) were observed with other cognitive measures. The direction of these effects suggested that longer daylengths were associated with slightly reduced cognitive speed and accuracy. To account for this, we included daylength as an adjustment factor in the analyses described in **Sections 2.4** (Comparison of light sensitivity and light exposure) and **Sections 2.5** (Comparison of cognitive performance and light exposure). Importantly, seasonal daylight duration did not alter the results, and this factor was not significant in multivariate regression models. Therefore, we conclude that seasonal variation in daylight duration does not confound the observed associations between ambient light exposure and cognitive performance.

2. There may be differences in the diurnal patterns of cognitive function between habitual midday nappers and non-habitual nappers. Has the study examined the potential moderating role of midday napping habits on the diurnal variation of cognitive performance?

Answer: We agree with the reviewer that napping may influence cognitive performance. While we did not directly compare habitual nappers and non-nappers, we recorded the time of the last awakening before each repeated assessment (**page 14** of the manuscript). Notably, time awake shows a strong association with both KSS scores and PVT reaction times, supporting its relevance as a covariate (**Figure 2**). Consequently, the “time awake” variable accounts for any naps, and this variable is included as a covariate in the analyses examining associations between recent light exposure and cognitive performance.

3. There is no mention of correction for multiple comparisons for the correlation analyses, even though a large number of bivariate relationships were tested (i.e., ≥ 3 correlations between light exposure and performance).

Answer: We agree with the reviewer that correction for multiple testing is a critical issue. In **Section 4.5** (Statistical Analysis), we have clarified that: "In each section, we applied the Bonferroni correction to adjust for multiple comparisons by setting the significance threshold as α divided by the number of predictors." In **Figures 2, 3, and 4**, heatmaps are used to display correlation results, with significant associations indicated by * and Bonferroni-corrected significant associations indicated by **.

4. Please add some discussions on the neural mechanisms in the Discussion section.

Answer: We thank the reviewer for this valuable suggestion. In response, we have added a paragraph to the **Discussion** section that summarizes potential neural mechanisms underlying our findings: “Overall, two key patterns emerged from our results: (1) recent light exposure was associated with cognitive effects consistent with increased arousal (e.g., heightened alertness and faster reaction times), and (2) a history of bright, stable daytime light exposure was linked to enhanced sustained attention in a visual search task. Both effects are likely initiated by activation of the ipRGC system, as laboratory studies have demonstrated a short-wavelength bias in the acute effects of light on arousal and cognitive function (consistent with melanopsin spectral sensitivity), alongside the established role of ipRGCs in regulating central clock function via the retinohypothalamic tract to the suprachiasmatic nucleus (SCN). Acute effects may involve not only ipRGC input to the SCN but also contributions from projections to other brain regions. For example, the locus coeruleus (LC) and its ascending reticular activating system have been implicated in mediating acute arousal, potentially through indirect input from the SCN. However, other ipRGC target regions may also contribute. Animal studies have identified several candidate pathways, though their relevance to humans remains less certain. Furthermore, improved attention under high-melanopic light suggests contributions of top-down attentional and executive mechanisms, likely mediated by ipRGC–thalamic–corticolimbic pathways influencing prefrontal cortical activity. For the longer-term effects, exposure to high-amplitude, early light patterns appears to strengthen both circadian robustness and sleep homeostasis, thereby supporting improved cognitive performance over time.”

5. Some of the references were outdated.

Answer: Upon reevaluating our references, we identified only two citations prior to the year 2000: Dijk, Duffy, & Czeisler (1992), “Circadian and sleep/wake dependent aspects of subjective alertness and cognitive performance” and Gouras & MacKay (1989), “Growth in amplitude of the human cone electroretinogram with light adaptation.” These studies are foundational, as the authors are pioneers in the field and their work provides essential context for explaining our findings. For this reason, we preferred to retain them. In addition, earlier references are limited to the original methodological papers introducing widely used instruments (KSS, PSQI, MCTQ), which we considered important to cite. To strengthen our reference list, we additionally included a recent systematic review and meta-analysis directly related to our findings: “Non-Image-Forming Effects of Daytime Electric Light Exposure in Humans: A Systematic Review and Meta-Analyses of Physiological, Cognitive, and Subjective Outcomes” (2025) - <https://doi.org/10.1080/15502724.2025.2493669> - and one controlled laboratory study: “Selective activation of ipRGC modulates working memory performance” (2025) - <https://doi.org/10.1371/journal.pone.0327349> -